# LiteGfm: A Lightweight Self-supervised Monocular Depth Estimation Framework for Artifacts Reduction via Guided Image Filtering

## ABSTRACT

Facing with two significant challenges for monocular depth estimation under a lightweight network, including the preservation of detail information and the artifact reduction of the predicted depth maps, this paper proposes a self-supervised monocular depth estimation framework, called LiteGfm. It contains a DepthNet with an Anti-Artifact Guided (AAG) module and a PoseNet. In the AAG module, a Guided Image Filtering with cross-detail masking is first designed to filter the input features of the decoder for preserving comprehensive detail information. Second, a filter kernel generator is proposed to decompose the Sobel operator along the vertical and horizontal axes for achieving cross-detail masking, which better captures the structure and edge feature for minimizing artifacts. Furthermore, a boundary-aware loss between the reconstructed and input images is presented to preserve high-frequency details for decreasing artifacts. Extensive experimental results demonstrate that LiteGfm under 1.9M parameters gets more optimal performance than state-of-the-art methods.

## CCS CONCEPTS

• **Computing methodologies** → **Scene understanding**; *Neural networks*.

## KEYWORDS

Monocular depth estimation, Guided image filter, Lightweight network

## 1 INTRODUCTION

Scene understanding plays a vital role in various tasks, such as autonomous driving, embodied navigation, and virtual scene construction. Particularly, the depth estimation task significantly enhances the performance of localization and segmentation. Recently, depth estimation based on images has become the mainstream method, incorporating both traditional approaches[1–3] and deep learning approaches, with the latter achieving superior performance. However, the general supervised deep learning frameworks of depth estimation depend on depth labels for training. It is prohibitively expensive to attain precise ground-truth depth measurements through sensors such as radar, depth cameras, and stereo cameras, hard to obtain

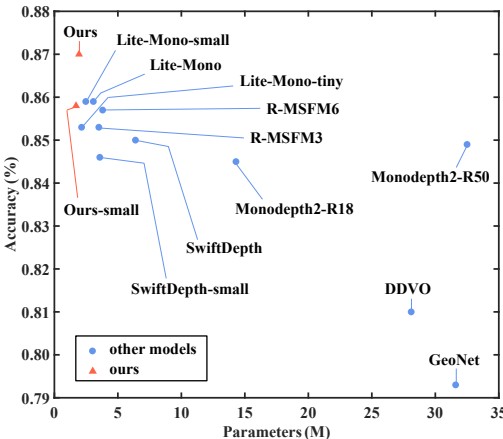

**Figure 1: Computational efficiency comparison. All results are tested on the KITTI dataset with a resolution of 192 × 640. The accuracy ($\delta_1$) and parameters of the representative models and LiteGfm are presented.**

across multiple scenes. Consequently, self-supervised depth estimation methods have gained popularity, with methods based on stereo images first to be proposed and displaying promising accuracy[4–6]. Despite this, they have limitations in terms of data collection with insufficient data. Therefore, by employing image reconstruction loss[7] and popular frameworks like ResNet[8] and VGG[9], monocular self-supervised depth estimation architectures have exhibited significant results in multitask learning[10–12]. However, mainstream network frameworks typically involve a large parameter and a considerable computational resources demand, which makes depth estimation on various edge devices challenging.

Nevertheless, the reduction in parameters leads to decreased fitting capacity, and existing lightweight methods frequently suffer from insufficient feature extraction. To address this, a dilated convolution[13] has been designed and implemented in lightweight frameworks to enhance feature extraction. It significantly increases the receptive field of filters, thereby enabling the network to capture features from a broader context range. However, the enlarged receptive field of dilated convolution results in a substantial loss of image details. Thus, often occurs in the production of depth maps with artifacts in depth prediction tasks.

First, to enhance feature detail information, a guided image filtering (GIF) module with a cross-detail masking filter is proposed. In contrast to the conventional approach of implementing filtering at the end of the network, the GIF module is placed at the decoder's input to conduct filtering procedures, effectively reducing the loss

of detailed information. Furthermore, the utilization of guided image filtering is favored, which transfers structural details from a guide image to the target image. The implementation of the GIF module significantly supplements the features of the decoder's input. Second, to eliminate edge artifacts, a lightweight filter kernel generator was designed for achieving cross-detail masking. This generator utilizes the Sobel operator in both vertical and horizontal directions, followed by pooling and point-wise multiplication operations, which enhance directional awareness features and effectively supplement structural and edge features. Finally, a boundary awareness loss is presented to eradicate detailed artifacts, which offers improved differentiation between details and artifacts in high-frequency regions and enhances the method's ability to eliminate detailed artifacts.

Hence, this paper proposes a lightweight but efficient approach for self-supervised monocular depth estimation. Based on the chosen backbone, a lightweight AAG module is incorporated in the decoder's input to fulfill the detailed information with a smooth gradient. The AAG module encompasses a filter kernel generator and GIF procedures. To avoid introducing extra parameters, we select the Sharr operator due to its ability to capture more comprehensive edge details, even weak edges. Moreover, a boundary-aware loss is employed to provide additional optimization of the reconstructed image for more detailed features. As illustrated in Figure 1, under identical experimental settings, our proposed LiteGfm demonstrates superior equilibrium between performance and complexity compared to the prevailing state-of-the-art method in monocular depth estimation. To summarize, this paper's contributions are:

- A novel lightweight self-supervised monocular depth estimation framework with a guided image filter architecture is proposed, called LiteGfm, which achieves the smallest model size and superior accuracy with extensive experiments on KITTI. Sufficient ablation studies confirm the effectiveness of different design choices.
- This paper presents a GIF module that applies guided image filtering at the decoder's input, significantly fulfilling the detailed information with reduced parameters for tackling the challenge of neglected detailed information.
- This paper develops a lightweight guided filter kernel generator achieving cross-detail masking that utilizes the Sobel operator to extract edge information, tackling the challenge of emerged edge artifacts. Additionally, boundary-aware loss contributes to different details and artifacts in high-frequency areas, which reduces the detailed artifacts in the predicted depth maps.

The subsequent sections are structured as follows: Section 2 provides an overview of relevant research studies. Section 3 presents the method in detail. Section 4 discusses the experimental results and analysis. Section 5 concludes the paper.

## 2 RELATED WORK

### 2.1 Self-supervised Monocular Depth Estimation

For monocular supervised methods, researchers consider it as a view synthesis task inspired by [14]. Then Zhou et al.[7] introduces an innovative approach with a distinct pose network to predict the 6-DoF pose between temporally adjacent frames, thereby substituting the established geometric constraints derived from stereo pairs. This has proved a lot of self-supervised monocular depth estimation. Based on it, certain researchers have proposed the incorporation of supplementary tasks, such as flow estimation[15, 16], semantic segmentation[17, 18], domain adaptation[19, 20], uncertainty estimation[21, 22], etc. Furthermore, improving loss can achieve favorable results without introducing supplementary tasks. The Monodepth2[10] incorporates the minimum re-projection loss to address occlusion challenges and employs automatic masking loss to eliminate moving objects at the same pace as the camera. Currently, Monodepth2 has emerged as the standard, and which self-supervised training strategy is also employed in our work.

In recent years, the emergence of edge devices has prompted researchers to focus on optimizing the balance between model complexity and accuracy. Some lightweight models are achieved by combining the new frameworks. HR-Depth[23] introduces a framework based on the MobileNetV3[24], yielding results similar to Monodepth2 with significantly fewer parameters. Similarly, SwiftDepth[25] creates an architecture with CNN and ViT[26], which demonstrates superior performance results. MViTDepth[27] also designs a novel architecture building upon MobileViT[28], utilizing it as a teacher model for knowledge distillation to compress the model and subsequently enhance the performance. Besides, XDistill[29]presents additional supervision for the DepthNet, facilitating cross-task knowledge distillation and enhancing prediction accuracy with small parameters. Moreover, certain approaches aim to balance model complexity and accuracy by enhancing feature extraction modules. R-MSFM[30], a lightweight model integrates the upsampling module of multi-scale feature modulation and parameter learning, enhancing depth estimation quality.

However, the reduction of parameters leads to decreased fitting capacity, and lightweight methods frequently suffer from insufficient feature extraction. A hybrid architecture Lite-Mono[31] proposes a Consecutive Dilated Convolutions (CDC) module to effectively extract rich multi-scale local features, which achieves comparable results with a reduction of approximately 80% parameters. But Lite-Mono utilizes the attention-based module, which adds a lot of parameters. This paper introduces guide image filtering and designs a lightweight filter kernel generator to decrease our model's complexity.

### 2.2 Guided Image Filter

Guided image filtering is a novel explicit image filter[32] that computes the filtering output by considering the content of a guidance image. It exhibits the nice property of edge-preserving smoothing while also being computationally efficient and exact. Through this, the utilization of images containing a greater amount of feature information as reference images enables the transfer of structural information to the desired target images. So some works try to utilize the guided image filter to refine object masks[33, 34]. Different from the classical guided image filtering, the deep learning-based designed guided image filtering approaches are proven. The recent deep joint filtering method designs two-branch convolution subnetworks to extract features from the guidance and the target

images[35]. Some combined the other mechanisms, such as deep attentional guided image filtering presenting a multiscale module to progressively generate the filtering result with the constructed kernels from coarse to fine with guided image filtering[36]. Correspondingly, a multiscale fusion strategy is introduced to reuse the intermediate results in the coarse-to-fine process. Guided image filtering has been implemented in various computer vision tasks, such as face recognition[37], semantic segmentation[38, 39], and image dehazing[40].

This paper refers to the concept of guided image filtering to transfer the structures of the guidance image to the filtering output. It solves the challenge of edge and detailed artifact elimination of the predicted depth maps.

## 3 METHODS

### 3.1 The Proposed Framework: LiteGfm

This paper proposes the LiteGfm framework, a lightweight self-supervised monocular depth estimation framework with a guided image filter module. As is shown in Figure 2, the LiteGfm framework consists of a DepthNet and a PoseNet. In the DepthNet, a target image $I^t \in \mathcal{R}^{H \times W \times 3}$ is fed into a convolution stem, which comprises a down-sampling layer (i.e. $3 \times 3$ convolutions with *stride = 2*.)and two $3 \times 3$ convolutions with *stride = 1*. Then, the features flow into the following three stages, and each stage consists of a down-sampling layer and a CDC module from Lite-Mono[31]. The CDC blocks in each stage have different dilation rates that are [1, 2, 3] for stage 1, [1, 2, 3] for stage 2, and [1, 2, 3, 2, 4, 6] for stage 3. This is utilized in the two sizes of the proposed LiteGfm.

In the decoder of the DepthNet, our proposed framework presents an AAG module to solve the problems of neglected detailed information and emerged artifacts. The AAG module consists of a lightweight guided kernel generator and GIF module, but effectively maintains the parameter only at 0.007 M, diminishing the model's complexity. Taking the first stage as an illustration, the feature map after the down-sampling layer and the pooled target image are both as input to generate a guided kernel. Thereafter, the guided kernels and each stage's output are input to the GIF module to achieve cross-detail masking. Then it outputs the feature maps with more detailed information and fewer artifacts. In the end, we utilize bilinear up-sampling to expand the spatial dimension. After each up-sampling block, a prediction block is followed to generate the inverse depth map at the assigned resolution. Likewise, the other two kernels can be generated by the feature among the other two stages. Our framework employs the identical PoseNet as[10, 31], which is input adjacent monocular frames to calculate the camera pose for achieving self-supervision learning.

Then this paper illustrates these present novel modules and the designed loss for training.

### 3.2 Guided Image Filtering

This part introduces the GIF module in the proposed AAG module. The challenge we intend to tackle is to output the depth map with the absence of details. As a result, we select the results of the down-sample layer in each stage as the target feature $I^t \in \mathcal{R}^{H \times W \times C^t}$, and the pooled source image with comprehensive feature information input as the guided feature $I^g \in \mathcal{R}^{H \times W \times C^g}$, where $H$, $W$,

and $C$ denote the height, width, and channels respectively. Combined with the filter kernels $W^k$ generated by $I^t$ and $I^g$, the input feature can be reconstructed with edges and gradients preserved by getting the prior information from the guided kernels. After passing through the below layers, the ultimate output of each stage $\left\{ F^i_{stage}, 0 \leq i < m \right\}$ will consequently drop some detailed information, which is crucial to the depth prediction. This GIF module utilizes detailed information of the object within the feature domain to enhance filtering outcomes.

As shown in Figure 2, this filter kernel's whole guidance process can be exposed to:

$$I_3 = Conv(F^3_{stage}), \qquad F_3 = GIF[I_3, W^k_3], \quad (1)$$

$$I_2 = Conv(Concat(F^2_{stage}, F^\uparrow_3)), \qquad F_2 = GIF[I_2, W^k_2], \quad (2)$$

$$I_1 = Conv(Concat(F^1_{stage}, F^\uparrow_2)), \qquad F_1 = GIF[I_1, W^k_1]. \quad (3)$$

where $F^i_{stage}$ denotes the final outputs of the $i$th stage; $I_i$ and $W^k_i$ denote the inputs of the $i$th process of the GIF module; $\uparrow$ is an upsampling operation. $GIF(\cdot, \cdot)$ is the guided image filter to guide the corresponding features. For feature $I_i$, the filter process is as follows :

$$F_i = \sum_{(h,w) \in (H,W)} W^k_i(h, w) I_i(h, w) \quad (4)$$

The filtered features $\{F_i, 0 \leq i < m\}$ are directed to capture added details, and are subsequently forwarded to the network to generate a superior depth map. The following elaborates on the guided image kernel generator and the loss designed for network training

### 3.3 Guided Image Filter Kernel Generator

The guided image filter is employed to upgrade the target image's structural details and eliminate the edge artifacts of depth maps. The proposed kernel generator is designed to produce kernels that incorporate guided constructive information through cross-detail masking.

As illustrated in Figure 3, it takes target feature $I^t$ and guided feature $I^g$ as the inputs. First, we utilize a two-branch network to completely extract detailed information crossly. Then combine the outputs of the two branches to constitute the kernel. Taking the horizontal direction as an illustration, we apply the Scharr operator in the horizontal direction, which can detect fainter edges with greater efficiency to enhance the comprehensive extraction of object edges of an image. Next, a pooling operation is conducted horizontally. The entire process can be calculated as:

$$F^t_{hor} = Avg(Sharr_{hor}(Conv(I^t))) \quad (5)$$

$$F^g_{hor} = Avg(Sharr_{hor}(Conv(I^g))) \quad (6)$$

where the $Avg$ encodes the feature $X_{hor} \in \mathcal{R}^{H \times W \times C}$ from horizontal directions with pooling kernels (H, 1), which is capable of extracting global context efficiently, can be presented as :

$$Avg(X_{hor}) = \frac{1}{W} \sum_{0 \leq i < W} X_{hor}(h, i) \quad (7)$$

Currently, the features collected by the guided and target features along the horizontal direction are subjected to max-pooling. Then

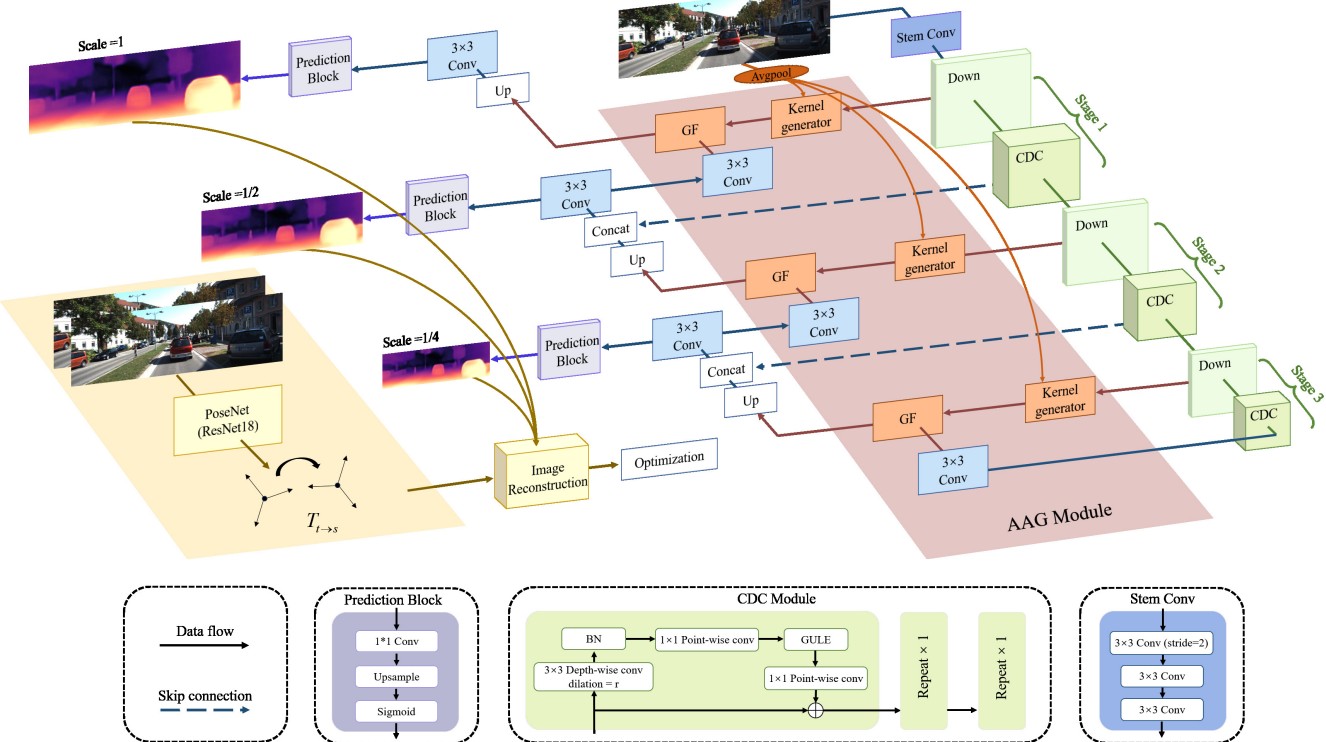

**Figure 2: Overview of the proposed LiteGfm. In the training stage, the target RGB image is input into the DepthNet. The encoder of the DepthNet uses dilated convolution to expand the receptive field, and the decoder generates a guided filter kernel to extract rich structural and edge information. Then, the target and adjacent frames are input to the PoseNet to obtain the relative pose of the camera. Finally, the loss is used to optimize the DepthNet and the PoseNet simultaneously. In the testing stage, only DepthNet is employed for depth prediction to obtain a depth map.**

we obtain the optimal edge information in the horizontal direction, denoted as $W_{hor}^k$. The process can be formulated as:

$$W_{hor}^k = Max(F_{hor}^t, F_{hor}^g) \tag{8}$$

Simultaneously, we also obtain an additional vertical branch outcome denoted as $W_{ver}^k$. It is assumed that:

$$W_{ver}^k = Max(F_{ver}^t, F_{ver}^g) \tag{9}$$

The final guided filter kernel, which achieves the cross-detail masking can be derived as:

$$W^k = W_{hor}^k \odot W_{ver}^k \tag{10}$$

where $W^k$ is the generated filter kernel; $\odot$ means element-wise multiplication.

## 3.4 Self-supervised Learning

Following [7], this work treats depth estimation as the task of image reconstruction. We first train a depth net, giving an RGB image $I_t$ as input, and output a per-pixel depth map $D_t$. Then, we train a pose net to estimate the relative camera pose $T_{t \to s}$ from temporally adjacent frames. So, warp $I_s$ into $I_t$ to generate the constructive image $I_{s \to t}$:

$$I_{s \to t} = I_s \langle proj(D_t, T_{t \to s}, K) \rangle \tag{11}$$

where $proj(\cdot)$ serves as the resulting 2D coordinates of the projected depths $D_t$ in $I_t$, $s \in [t-1, t+1]$, and $\langle \cdot \rangle$ is the sampling operator. Finally, the image reconstruction loss is used to optimize our network.

*3.4.1 Image Reconstruction Loss.* The per-pixel reprojection loss is defined as:

$$\ell_p(I_s, I_t) = \min_{I_s \in [-1,1]} \ell_p(I_{s \to t}, I_t) \tag{12}$$

where $\ell_p$ is the photometric loss defined as:

$$\ell_p(I_{s \to t}, I_t) = \alpha \frac{1 - SSIM(I_{s \to t}, I_t)}{2} + (1 - \alpha) \|I_{s \to t} - I_t\| \tag{13}$$

where SSIM is the structural similarity index measure[41] and $\alpha$ is set to 0.85[10].

Auto-masking is applied to remove moving pixels where no relative camera motion is observed[10]:

$$\mu = \min_{I_s \in [-1,1]} \ell_p(I_s, I_t) > \min_{I_s \in [-1,1]} \ell_p(I_{s \to t}, I_t) \tag{14}$$

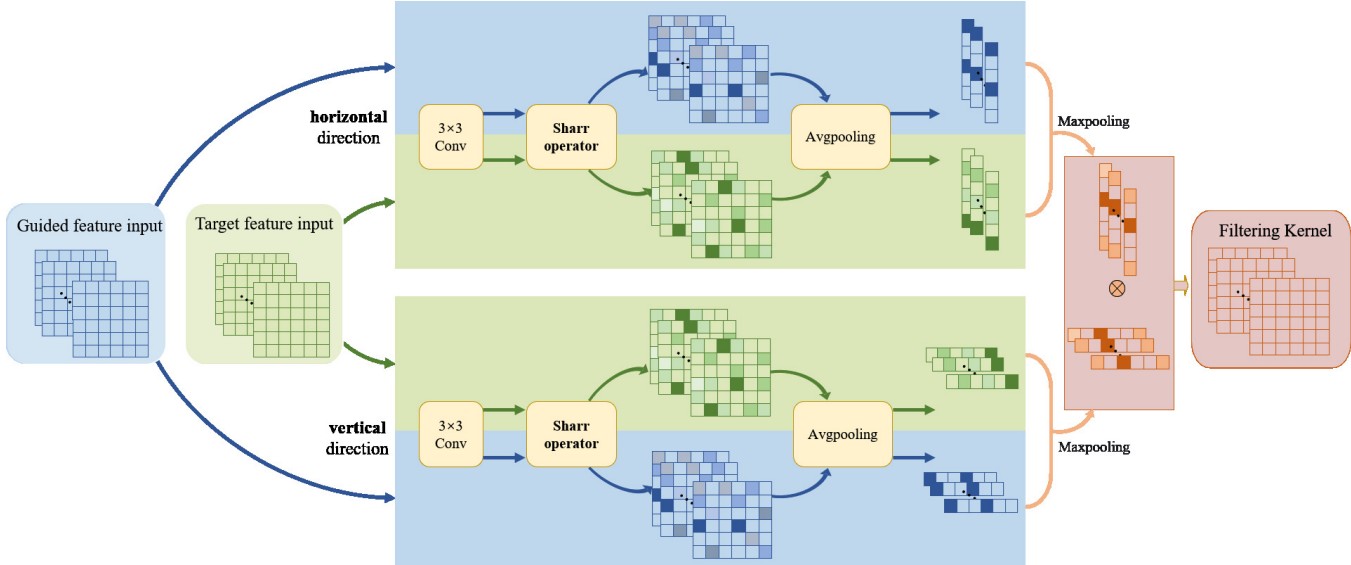

**Figure 3: The structure of the proposed kernel generator. The target and guided features are input to a two-branch network for calculating a guided image filter kernel.**

Therefore, the image reconstruction loss can be expressed as:

$$\ell_r(I_{s\to t}, I_t) = \mu \cdot \ell_p(I_s, I_t) \tag{15}$$

*3.4.2 Edge-aware Smoothness Loss.* Following [10, 42], the edge-aware smoothness regularization loss is applied to enhance the depth estimation at object boundaries, followed by:

$$\ell_{smooth} = \left|\partial_x d_t^*\right| e^{-\left|\partial_x I_t\right|} + \left|\partial_x d_t^*\right| e^{\left|\partial_y I_t\right|} \tag{16}$$

where $d_x^* = d_t/\hat{d}_t$ is the mean-normalized disparity.

*3.4.3 Boundary-aware Loss.* Due to the equal treatment of all pixels, the mentioned approach of optimizing the loss at the pixel level fails to effectively retain high-frequency structural details and causes detailed artifacts in a depth map. To address this issue and encourage the network to prioritize high-frequency components, a boundary-aware loss is introduced. This loss aims to improve the model's capability to generate sharper boundaries and intricate details. Followed by [43], the boundary-aware loss is defined as:

$$\ell_{ba} = \|M \odot I_t - M \odot I_{s\to t}\|_1 \tag{17}$$

where we define the $M$ as the boundary mask:

$$M = (\nabla_x I^t - \nabla_x I_{s\to t}) \odot (\nabla_y I^t - \nabla_y I_{s\to t}) \tag{18}$$

where $\odot$ denotes the elementwise multiplication and $\nabla$ is the Sobel operator to detect the boundary information. With these three losses, the total loss is formulated as:

$$\ell = \lambda_1 \ell_p + \lambda_2 \ell_{smooth} + \lambda_3 \ell_{ba} \tag{19}$$

where $\lambda_1, \lambda_2, \lambda_3$ are set as 1, $1e^{-3}$, 0.02 respectively. We need to enhance the edge information without compromising the accuracy of the prediction, while the consistency loss prevails, and $\lambda_2$ is set as [10].

## 4 EXPERIMENTS

In this section, a thorough explanation of the implementation of our framework is provided. Then, we evaluate the proposed LiteGfm and LiteGfm-small frameworks on KITTI datasets. Next, comprehensive ablation experiments are conducted to verify the effect of each contribution. Finally, visual experiments demonstrate the impact of these components within the framework individually.

### 4.1 Implementation Details

*4.1.1 Dataset.* The KITTI dataset[44] is used to train and evaluate the latest SOTA algorithms. The Eigen split[14] which has 39,810 monocular triplets for training, 4,424 images for validation, and 697 for testing is applied. The input image resolutions are set to 192×640. Similar to MonoDepth2[10], the predicted depth is confined in the range of [0, 80] m.

*4.1.2 Hyperparameters.* This work is implemented in PyTorch and trained on NVIDIA GeForce RTX 3090 with a batch size of 12. The optimizer is AdamW with the weight decay set to $1e^{-2}$. All experiments involving the compared methods are trained from scratch. We set the initial learning rate to $5e^{-4}$ and change it to $1e^{-4}$ from 31 epochs for fine-tuning. The network is trained for 60 epochs, which takes about 20 hours.

*4.1.3 Evaluation metrics.* The performance of the proposed framework is reported by the standard metrics proposed from[49], consisting of absolute relative difference (Abs Rel), square-related difference (Sq Rel), root mean square error (RMSE), RMSE log, $\delta 1 < 1.25$, $\delta 2 < 1.25^2$, $\delta 3 < 1.25^3$.

### 4.2 Depth Estimation Results

Table 1 displays the experiment results of LiteGfm and some representative methods with model sizes lower than 35 M on the KITTI

**Table 1: The quantitative results of LiteGfm with some recent representative methods on the Eigen split[14] of the KITTI dataset[44]. The best results are presented in bold for each category, with the second-best results underlined.**

| Methods | Abs Rel | Sq Rel | RMSE | RMSE log | $\delta 1$ | $\delta 2$ | $\delta 3$ | Param. |
|---|---|---|---|---|---|---|---|---|
| Zhou[45] | 0.183 | 1.595 | 6.709 | 0.270 | 0.734 | 0.902 | 0.959 | 31.6M |
| GeoNet[46] | 0.155 | 1.296 | 5.857 | 0.233 | 0.793 | 0.931 | 0.973 | 31.6M |
| DDVO[47] | 0.151 | 1.257 | 5.583 | 0.228 | 0.810 | 0.936 | 0.974 | 28.1M |
| EPC++[48] | 0.141 | 1.029 | 5.350 | 0.216 | 0.816 | 0.941 | 0.976 | 33.2M |
| MonoDepth2-ResNet18[10] | 0.132 | 1.044 | 5.142 | 0.210 | 0.845 | 0.948 | 0.977 | 14.3M |
| MonoDpeth2-ResNet50[10] | 0.131 | 1.023 | 5.064 | 0.206 | 0.849 | 0.951 | 0.979 | 32.5M |
| R-MSFM3[30] | 0.128 | 0.965 | 5.019 | 0.207 | 0.853 | 0.951 | 0.977 | 3.5M |
| R-MSFM6[30] | 0.126 | 0.944 | 4.981 | 0.204 | 0.857 | 0.952 | 0.978 | 3.8M |
| Lite-Mono-tiny[31] | 0.125 | 0.935 | 4.986 | 0.204 | 0.853 | 0.950 | 0.978 | 2.1M |
| Lite-Mono-small[31] | 0.123 | 0.919 | 4.926 | 0.202 | 0.859 | 0.951 | 0.977 | 2.5M |
| Lite-Mono[31] | 0.121 | 0.876 | 4.918 | 0.199 | 0.859 | 0.953 | 0.980 | 3.1M |
| SwiftDepth-small[25] | 0.132 | 1.040 | 5.148 | 0.210 | 0.846 | 0.948 | 0.976 | 3.6M |
| SwiftDepth[25] | 0.128 | 1.020 | 5.093 | 0.205 | 0.850 | 0.951 | 0.978 | 6.4M |
| LiteGfm-small (ours) | 0.123 | 0.924 | 4.922 | 0.199 | 0.858 | 0.953 | 0.980 | **1.7M** |
| LiteGfm (ours) | **0.117** | **0.871** | **4.797** | **0.194** | **0.870** | **0.957** | **0.981** | 1.9M |

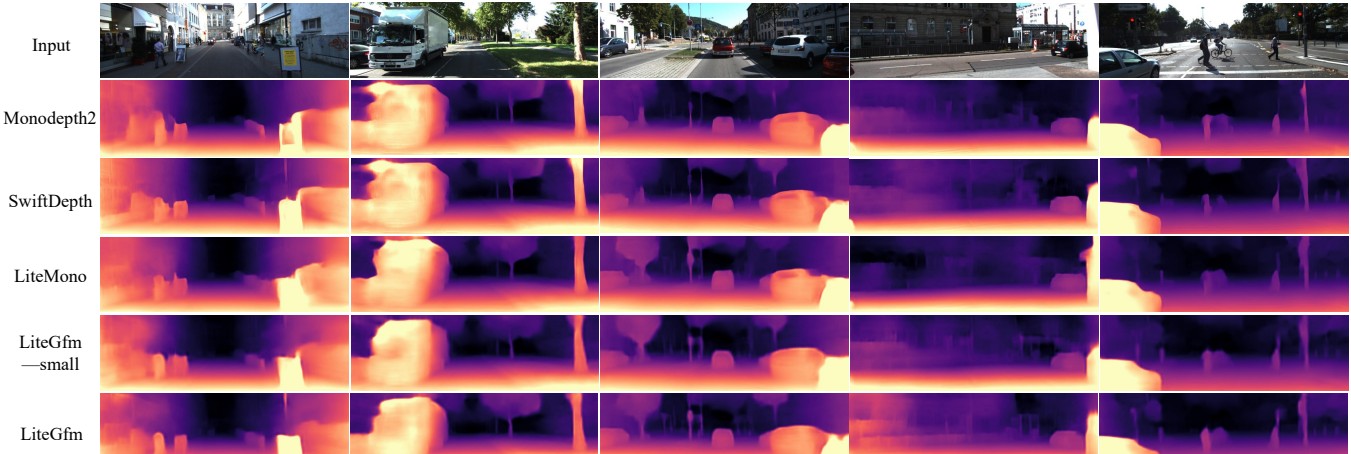

**Figure 4: Visualization of LiteGfm and some methods[10, 25, 31] on the Eigen split[14] of the KITTI dataset[44].**

dataset[44]. All of the results are without pre-training on ImageNet. Compared to the standard model Monodepth2 of the ResNet50 version and the current advanced lightweight model Lite-Mono, the LiteGfm performs superior with the minimal parameter (1.9 M). The small-size LitGfm, with the lowest parameters (1.7 M), also achieves satisfactory results among all small-size models, even most of the full-size models.

Moreover, Figure 4 shows the quantitative comparison of depth maps among these lightweight models. LiteGfm decreases the edge artifacts through the cross-detail masking, which objects in the depth maps have a clearer edge, such as the billboard with a regular shape in column 1 and the vehicles of different sizes and colors in columns 2 and 3. Besides, more detailed artifacts are presented in Lite-Mono and SwiftDepth, yet less on LiteGfm which has a remarkable performance on the shady wall with complicated structure in column 4 and distant traffic lights or slender poles in columns 3 and

5. Furthermore, with the boundary-aware loss, LiteGfm displays a more stable prediction, which can be verified from the great prediction of trees and pedestrians with light changes in columns 2 and 5. In a word, LiteGfm achieves satisfactory results in decreasing the edge and detailed artifacts.

### 4.3 Complexity and Speed Evaluation.

The performance of the proposed framework LiteGfm is evaluated in terms of model complexity and inference speed. As shown in Table 2, our LiteGfm has the lowest model parameters, which is decreased by 87% compared to Monodepth2[10]. Coming to the decoder part, our filtering module evitably increases the FLOPs and inference time. Furthermore, the comparison of computational efficiency is depicted in Figure 1.

**Table 2: Model complexity and speed evaluation. Params denote the number of parameters. FLOPs are floating points of operations. Speed is inference time.**

| Methods | Encoder | | | Decoder | | | Full Model | | |
|---|---|---|---|---|---|---|---|---|---|
| | Params (M) | FLOPs (G) | speed (ms) | Params (M) | FLOPs (G) | speed (ms) | Params (M) | FLOPs (G) | speed (ms) |
| MonoDepth2 | 11.2 | 4.5 | 0.8 | 3.1 | 3.5 | 0.9 | 14.3 | 8.0 | 1.7 |
| R-MSFM3 | 0.7 | 2.4 | 0.2 | 2.8 | 14.1 | 3.4 | 3.5 | 16.5 | 3.6 |
| R-MSFM6 | 0.7 | 2.4 | 0.2 | 3.1 | 28.8 | 5.6 | 3.8 | 31.2 | 5.8 |
| MViT-Depth-tiny | 1.0 | 0.7 | 1.5 | 0.8 | 0.8 | 0.3 | 1.8 | 1.5 | 1.8 |
| MViT-Depth-small | 1.9 | 1.8 | 1.9 | 0.9 | 1.0 | 0.4 | 2.8 | 2.8 | 2.3 |
| MViT-Depth | 5.0 | 3.6 | 2.2 | 1.3 | 1.1 | 0.4 | 6.3 | 4.7 | 2.6 |
| Lite-Mono-tiny | 2.0 | 2.4 | 1.6 | 0.2 | 0.7 | 0.2 | 2.5 | 4.8 | 1.8 |
| Lite-Mono-small | 2.3 | 4.1 | 2.0 | 0.2 | 0.7 | 0.2 | 2.5 | 4.8 | 2.2 |
| Lite-Mono | 2.9 | 4.4 | 2.1 | 0.2 | 0.7 | 0.2 | 3.1 | 5.1 | 2.3 |
| SwiftDepth-small | 3.0 | 1.5 | 1.7 | 0.6 | 2.1 | 0.2 | 3.6 | 3.6 | 1.9 |
| SwiftDepth | 5.6 | 2.4 | 2.2 | 0.8 | 2.5 | 0.2 | 6.4 | 4.9 | 2.4 |
| LiteGfm-small | 1.5 | 1.9 | 1.2 | 0.2 | 0.5 | 0.9 | 1.7 | 2.4 | 2.1 |
| LiteGfm | 1.7 | 3.3 | 1.7 | 0.2 | 0.7 | 0.9 | 1.9 | 4.0 | 2.6 |

## 4.4 Ablation Experiments

In this part, we verify the components of our model contribute to the overall performance. Above all, as shown in Figure 2, the AAG module mainly consists of the kernel generator and the guided filtering process. For the kernel generator, our model utilizes the current method which is two branches with the Scharr operator. Before this, we tried other kernel generation methods. We differentiate them with *new filter kernels* and *old filter kernels*. For the guided filtering process, we select the guided object in a strategy that maximizes the complement of structure and edge information, i.e. the features in the decoder. Additionally, the features output by each stage in the encoder are tried to filter and then feed the results into the decoder. This paper use *in encoder* and *in decoder* to distinguish. Besides, we execute boundary-aware loss $\ell_{ba}$ to train our model to avoid the artifacts of depth maps. In Table 3, the five variants of the model are presented the following:

- **Model 1**, which is the backbone without guided image filter, BA loss.
- **Model 2**, which is the backbone trained with only BA loss.
- **Model 3**, which uses the *old filter kernel* to guide the encoder's features, is trained with BA loss.
- **Model 4**, which uses the *old filter kernel* to guide the decoder's features, is trained with BA loss.
- **LiteGfm**, which uses the *new filter kernel* to guide the decoder's features, is trained with BA loss.

The quantitative results of ablation are listed in Table 3. Model 1 exhibits the worst performance in the absence of any of our contributions. The performance of the proposed LiteGfm surpasses that of the ablated models from the metrics results. Furthermore, every component proposed in our model has the potential to enhance the network performance substantially. The following gives a detailed analysis and some visualization results of these components.

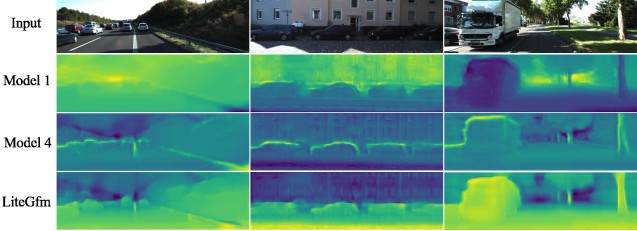

**Figure 5: Comparison of feature maps filtered by different guided filtering kernels.**

## 4.5 Visualization of the Different Guided Filter Kernels

This part shows the experimental results of different filter kernels designed in the experimental process. As shown in Table 3, compared with the model without a guided image filter, both models with a guided image filter in the decoder only increase the number of parameters by 0.007 M. However, this slight increase in parameters leads to a significant improvement in the model's accuracy. An initial method to create the guided image filter kernels involves iteratively subtracting the two sets of features of the input generator and subsequently establishing a cross-connection to supplement the information. However, it is noted that filters generated through this approach may disregard a significant portion of faint edge details. Consequently, in light of the artifacts stemming from the absence of structural and edge information, a dual-branch structure is employed to capture more subtle edges using the Scharr operator.

Figure 5 displays the depth maps produced in the absence of the guided image filter (i.e. Model 1) alongside those generated with two distinct guided image filters (i.e. Model 4 and LiteGfm), and the resultant feature map following decoder filtering. Model 1 shows the feature map outputs of the decoder at the corresponding stage. Notably, without a guided image filter, the baseline fails to perceive the scene structure, especially in regions with few substances to

Table 3: Results of ablation experiments in the five models.

| | Loss | filter kernels | guided object | Params (M) | Abs Rel | Sq Rel | RMSE | RMSE log | $\delta 1$ | $\delta 2$ | $\delta 3$ |
|---|---|---|---|---|---|---|---|---|---|---|---|
| Model1 | \ | \ | \ | 1.936 | 0.121 | 0.948 | 4.925 | 0.200 | 0.863 | 0.954 | 0.979 |
| Model2 | √ | \ | \ | 1.936 | 0.119 | 0.945 | 4.915 | 0.198 | 0.865 | 0.955 | 0.979 |
| Model3 | √ | old | in encoder | 1.957 | 0.120 | 0.933 | 4.910 | 0.198 | 0.869 | 0.956 | 0.980 |
| Model4 | √ | old | in decoder | 1.943 | 0.120 | 0.927 | 4.908 | 0.197 | 0.866 | 0.956 | 0.980 |
| LiteGfm | √ | new | in decoder | 1.943 | 0.117 | 0.871 | 4.797 | 0.194 | 0.870 | 0.958 | 0.981 |

find clues. A designed guided image filter can restore the depth of these regions with better accuracy and fewer edge and detailed artifacts.

## 4.6 Benefits of Guided Object

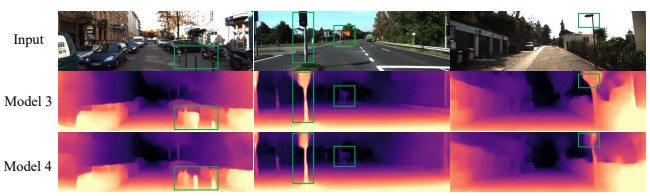

Figure 6: Comparison of depth maps of different guided objects.

Model 3 and Model 4 use a consistent filter kernel to process the characteristics of distinct stages in the encoder and decoder respectively.

From Table 3, the quantization results for Model 3 are significantly poorer than those for Model 4. An extra 0.016M parameter is needed for Model 3 to synchronize the filter kernel and features. Therefore, choosing to filter the features in the decoder facilitates the efficient incorporation of spatial details, mitigating redundant extraction of structural information, and minimizing parameter count.

Figure 6 shows the comparison of prediction depth maps with different guided objects. Compared to the filtering in the encoder, the results of filtering in the decoder provide the depth prediction with more details, including the road signs, poles, and billboards (the box area in Figure 6). Meanwhile, filtering in the decoder presents an effective ability to decrease the artifacts, particularly the objects near the edge of the image.

## 4.7 Benefits of BA loss

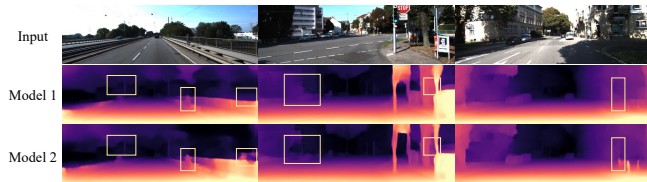

Figure 7: Comparison of depth maps with or without $\ell_{ba}$.

It can be challenging to differentiate between artifacts and actual details in parts of the image with high frequencies, leading to inaccuracies in reconstructing the image[50]. By utilizing $\ell_{ba}$, the network is prompted to prioritize high-frequency elements, enhancing the model's ability to distinguish between genuine edges and artifacts. Consequently, the depth map produced by our model exhibits a more refined structure with reduced instances of detailed artifacts.

Table 3 demonstrates that the incorporation of $\ell_{ba}$ significantly improves the performance from the quantitative results of Model 1 and Model 2. Meanwhile, Figure 7 displays the visualization results of Model 1 and Model 2 for comparison. When the light changes greatly in the image (the box area), the introduction of $\ell_{ba}$ can capture more details for the prominence of detailed artifacts.

## 5 CONCLUSIONS

This paper proposes a lightweight self-supervised monocular depth estimation method called LiteGfm to tackle the challenges that are the preservation of detailed information and the artifact reduction of the predicted depth maps. In the proposed architecture, an AAG module involving a GIF module with cross-detail masking and a filter kernel generator is presented. The GIF module uses the cross-detail masking filter to execute the input features of the decoder, which preserves comprehensive detail information. Additionally, a filter kernel generator is proposed to decompose the Sobel operator along the vertical and horizontal axes for achieving cross-detail masking, which is devoted to decreasing the edge artifacts. For minimizing detailed artifacts, a boundary-aware loss between the reconstructed and input images is presented to preserve high-frequency details. Extensive experiments on the Kitti dataset demonstrate that LiteGfm effectively reduces the number of parameters and achieves superior performance.

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
