# OpenReview forum: "LiteGfm: A Lightweight Self-supervised Monocular Depth Estimation Framework for Artifacts Reduction via Guided Image Filtering"
_acmmm.org/ACMMM/2024/Conference — MM2024 Poster_

### Official Review · Reviewer_WETt · 2024-05-22

**Rating:** 2
**Confidence:** 3

**Summary:**

In this paper, a lightweight self-supervised monocular depth estimation framework with a guided image filter architecture is proposed. The qualitative and quantitative results seem impressive, but the organization of the paper needs significant improvement.

**Strengths:**

1. The model proposed by the authors achieves impressive performance with a relatively small number of parameters.
2. Ablation experiments demonstrate the effectiveness of each design.

**Limitations:**

1. The article lacks clarity in its contributions, as the role and significance of the AAG module proposed by the authors are not reflected in the summary of the three main contributions in the article.
2. In Figure 2, the GF module in the AAG module is ambiguous; it is unclear whether it refers to the GIF module proposed by the authors.
3. The authors did not compare their method with lightweight state-of-the-art methods, such as Lite-HR-Depth (AAAI 2021), MViTDepth (RA-L 2023), etc.
4. The experimental dataset is limited, and it would be preferable to provide test results on the Make3D dataset, similar to previous methods.

**Suitability:**

3

---

### Official Review · Reviewer_X8hB · 2024-05-24

**Rating:** 6
**Confidence:** 3

**Summary:**

This paper is about to use a lightweight but efficient approach for self-supervised monocular depth estimation.

**Strengths:**

This paper proposes a novel lightweight self-supervised monocular depth estimation framework to achieve the smallest model size and superior accuracy. The proposed LiteGfm can preserve detailed information and reduce the artifacts on the predicted depth maps. The authors conducted extensive experiments on the KITTI dataset to demonstrate the performance of the proposed framework and the results are impressive.

**Limitations:**

The performance of the proposed framework is more optimal than the other state-of-the-art methods on the KITTI dataset. However, the authors are expected to show its performance on other realistic situations, like in a hazy day. Can the proposed LiteGfm be used to predict a depth map for a secen with strong scattering and how about its quality?

**Suitability:**

3

---

### Official Review · Reviewer_JHkS · 2024-05-25

**Rating:** 2
**Confidence:** 3

**Summary:**

This paper proposed a self-supervised monocular depth estimation framework, which consists of a DepthNet and a PoseNet.

**Strengths:**

1. The proposed method achieves better results than the competing methods in terms of estimation accuracy and model size.
2. Ablation studies are conducted to validate the effectiveness of each proposed module.

**Limitations:**

1. The logic of this paper is not clear somewhere. For example, the last two paragraphs in Introduction can be unified elegantly. Besides, there are some redundant descriptions throughout the paper. In Fig.2, the text of “GF” in the AAG module may mislead readers easily. It is better to replace “GF” with “GIF” for easier understanding.
2. For all the 9 competing methods, only three were published after 2019 (one in 2021 and two in 2023). Thus, I think the comparison with other methods fails to demonstrate the superior performance of the proposed method.
3. The runtime of the proposed method is longer than most of the competing methods.
4. The boundary-aware loss is proposed in another work. This work just simply uses it to train the network. Therefore, it cannot be seen as a kind of contribution. Furthermore, the boundary-aware loss should be removed for model1, model2, model3, and model4 to demonstrate the effectiveness of each proposed module in ablation studies.
5. All the experiments are conducted only on a dataset. More datasets should be used.

**Suitability:**

3

---

### Official Review · Reviewer_bm3V · 2024-05-30

**Rating:** 6
**Confidence:** 3

**Summary:**

This paper introduces LiteGfm, a novel lightweight self-supervised monocular depth estimation framework with a guided image filter architecture, which achieves the smallest model size and superior accuracy with extensive experiments on KITTI. Specifically, the AAG module is designed to effectively preserve detailed information and reduce artifact, a common problem faced by many depth estimation methods. Ablation studies confirm the effectiveness of the design choices.

**Strengths:**

-	A novel lightweight network architecture, LiteGfm is carefully designed to reduce artefacts in monocular depth estimation. Particularly impressive is the lightweight Anti-Artifact Guided (AAG) module which uses guided Image filtering and boundary-aware loss to effectively preserve details and reduce artefacts. This module can potentially be adopted for many other image reconstruction tasks.
-	Additionally, adopting a PoseNet for self-supervised learning is innovative and effective.
-	Ablation studies reinforce the role of the guided filtering and boundary-aware loss in improving the performance of the mode.
-	Experimental results demonstrates that the proposed approach can achieve state-of-the-art results with a mere 1.9M parameters.
-	The paper is generally well-written, well-organized and easily comprehensible.

**Limitations:**

-	From Table 2, it can be observed that while LiteGfm has the leased parameters, its FLOPs is higher than a couple of other heavier models. Some discussion on this observation would be interesting.
-	Evaluation is only performed on a single benchmark dataset. The results would ne more convincing if it demonstrates comparable performance on other benchmark datasets; e.g. NYUv2 dataset which is an indoor dataset to investigate its generalizability
-	A careful proofread is required as there are some minor typo/mistakes in the paper, e.g. Line 628: “quantitative comparison” should be “qualitative comparison”?

**Suitability:**

2

---

### Official Review · Reviewer_AgYr · 2024-05-31

**Rating:** 4
**Confidence:** 2

**Summary:**

The author proposes a lightweight framework to achieve the monocular depth estimation. The framework designs a novel AAM module to enhance the useful information and utilizes the Sharr operator to avoid extra parameters. The experimental results are sufficient enough to prove the effectiveness of proposed method.

**Strengths:**

This paper is easy to follow. The experiments are enough and convincing.
The proposed method can make a good trade-off between performance and complexity.
The author provides a comprehensive experimental result to evaluate the effectiveness of proposed module.

**Limitations:**

The quality of figures is bad, where there exists obvious distortion for the zoom-in framework.  The author should provide a more clear figure.

There exist some capital error mistakes, such as 701 'speed'->'Speed' and key words parts.

Since many formulas are defined in this paper, the reviewer suggests adding a table including the definitions of all the parameters of different modules for clear clarification.

**Suitability:**

3

---

### Meta-Review · Area_Chair_WXE6 · 2024-07-04

**Recommendation:** Accept (Poster)
**Confidence:** 4

**Metareview:**

This paper introduces LiteGfm, a compact and accurate self-supervised monocular depth estimation framework. LiteGfm includes DepthNet with an Anti-Artifact Guided (AAG) module and a PoseNet. It demonstrates excellent performance and small model size in tests on the KITTI dataset. The AAG module helps maintain detail and reduce artifacts, a common issue in depth estimation. Ablation studies validate the design's effectiveness. The paper is easy to follow. Most of the issues raised by the reviewers were adequately answered by the reviewers in the rebuttal. A few points that still need the authors' attention include conducting experimental evaluation on more datasets as all reported experiments are performed on a single dataset. Contributions must be revisited as pointed out by JHkS.